# Characterization of Extractable and Non-Extractable Phenols and Betalains in Berrycactus (*Myrtillocactus geometrizans*) and Its Chemoprotective Effect in Early Stage of Colon Cancer In Vivo

**DOI:** 10.3390/antiox13091112

**Published:** 2024-09-14

**Authors:** Rosa Iris Godínez-Santillán, Aarón Kuri-García, Iza Fernanda Ramírez-Pérez, María Guadalupe Herrera-Hernández, Santiaga Marisela Ahumada-Solórzano, Salvador Horacio Guzmán-Maldonado, Haydé Azeneth Vergara-Castañeda

**Affiliations:** 1Center for Advanced Biomedical Research, School of Medicine, Autonomous University of Queretaro, Campus Aeropuerto Carretera a Chichimequillas S/N, Ejido Bolaños, Querétaro 76140, Querétaro, Mexico; lic.nutricion.rosairis@gmail.com; 2Department of Cell and Molecular Biology, School of Natural Sciences, Universidad Autónoma de Querétaro, Querétaro 76230, Querétaro, Mexico; aaron.kuri@uaq.mx; 3School of Chemistry, Universidad Autónoma de Querétaro, Cerro de las Campanas, Querétaro 76076, Querétaro, Mexico; iza.perez@uaq.edu.mx; 4Unidad de Biotecnología, Campo Experimental Bajío, Instituto Nacional de Investigaciones Forestales, Agrícolas y Pecuarias (INIFAP), Celaya 38110, Guanajuato, Mexico; herrera.guadalupe@inifap.gob.mx; 5Investigación Interdisciplinaria en Biomedicina, School of Natural Sciences, Universidad Autónoma de Querétaro, Querétaro 76230, Querétaro, Mexico; santiaga.marisela.ahumada@uaq.mx

**Keywords:** berrycactus, extractable and non-extractable polyphenols, colon cancer, aberrant crypt foci, β-glucuronidase, short-chain fatty acids

## Abstract

This research identified the bioactive compounds and antioxidant capacity of the extractable (EP) and non-extractable (NEP) polyphenol fractions of berrycactus (BC). Additionally, the effects of BC and its residue (BCR) on preventing AOM/DSS-induced early colon carcinogenesis were evaluated in vivo. Male Sprague Dawley rats were randomly assigned to six groups (n = 12/group): healthy control (C), AOM/DSS, BC, BCR, BC+AOM/DSS, and BCR+AOM/DSS. NEP was obtained through acid hydrolysis using H_2_SO_4_ and HCl (1 M or 4 M). The HCl-NEP fraction exhibited the highest total phenolic and flavonoid content, while condensed tannins were more abundant in the H_2_SO_4_-NEP fraction. A total of 33 polyphenols were identified by UPLC-QTOF-MSE in both EP and NEP, some of which were novel to BC. Both NEP hydrolysates demonstrated significant total antioxidant capacity (TEAC), with HCl-NEP exhibiting the highest ORAC values. The BC+AOM/DSS and BCR+AOM/DSS groups exhibited fewer aberrant crypt foci (*p* < 0.05), reduced colonic epithelial injury, and presented lower fecal β-glucuronidase activity, when compared to AOM/DSS group. No differences in butyric acid concentrations were observed between groups. This study presents novel bioactive compounds in EP and NEP from BC that contribute to chemopreventive effects in early colon carcinogenesis, while reducing fecal β-glucuronidase activity and preserving colonic mucosal integrity.

## 1. Introduction

The term “non-extractable polyphenols (NEP)” was described in 2012 and refers to the polyphenols that remain bound to the polysaccharides of a food mainly through covalent and non-covalent bonds and that elude quantification by standard water extractions. They are not bioavailable in the small intestine and enter the large intestine where they are released by the action of the microbiota, and can thus exert important biological activities. The study of NEP is facilitated when samples are treated with acids (HCl, H_2_SO_4_) that allow the release of large amounts of polyphenolic compounds [1]. On the other hand, dietary fiber is a macromolecule that has been shown to draw non-extractable polyphenols into the colon [2]. One fruit rich in dietary fiber is berrycactus, locally called “garambullo”, a fruit endemic to Mexico that has received special attention in recent years due to the diversity of its phytochemical compounds [3,4]. Sánchez-Recillas et al. [5] analyzed the metabolites of berrycactus before and after an in vitro digestion process in the small intestine. The result was the identification of polyphenols from the families of flavanones, isofavones, flavonols, hydroxycinnamic acids, hydroxybenzoic acids, and betalains, mainly in the non-digestible fraction of berrycactus, suggesting a certain protection of these compounds by dietary fiber so that they can reach the colon. In addition, Sánchez-Recillas et al. [6] investigated the metabolites produced in the colon after in vitro fermentation of non-digested extract from berrycactus after an in vitro digestion process and identified 50 metabolites, such as benzene, indoles, phenols, and fatty acids, mainly short-chain fatty acids. Some of these metabolites are associated with protection against diseases such as colon cancer, through different mechanisms [7,8]. Colon cancer (CC) is a global public health problem; it is estimated that in 2020 there were more than 1.9 million new cases of this type of cancer worldwide and 930,000 deaths due to this pathology. Effective prevention, timely diagnosis, detection and removal of precancerous lesions, and the choice of appropriate therapies are crucial factors for patient survival [9,10]. Within the prevention of CC, there are modifiable factors such as diet, with a high intake of red and processed meat and high alcohol consumption increasing the risk, while foods with fiber and phytochemicals are considered protective factors [11]. Against this background, it could be hypothesized that the consumption of berrycactus could prevent the development of lesions representative of the first stages of colon carcinogenesis due to its nutraceutical properties. Therefore, the aim of the present study was the phytochemical characterization of extractable polyphenols (EP), NEP, and betalains, as well as the antioxidant capacity and the chemopreventive evaluation of berrycactus fruit in the initial phase of the carcinogenic process of the colon induced in an animal model.

## 2. Materials and Methods

### 2.1. Fruit Sample

Berrycactus was harvested in Guanajuato, Mexico, in June 2020. The fruits were sorted according to the stage of maturity (purple) [12]. After harvesting, they were washed and freeze-dried (Labconco Stoppering Tray Dryer 775013, Kansas City, MO, USA). They were then ground in an electric grinder (nutribullet NB603DG, Mexico City, México) and the powder was stored in a freezer (REVCO last II, Columbus, OH, USA) at −80 °C until analysis.

### 2.2. Extractable Phenolics

Freeze-dried samples were extracted with aqueous methanol (methanol-water 30:70 *v*/*v*). A digital orbital stirrer (model OM10E, brand OVAN, Burladingen, Germany) was used at 8000 rpm at room temperature for 20 min, then an ultrasonic bath was used at room temperature for 20 min, and finally the samples were centrifuged at 5000 rpm for 10 min [13]. The supernatants (extracts) were used for further analysis. The berrycactus residues (BCR) were reserved for the extraction of NEPs.

### 2.3. Non-Extractable Phenolics

BCR were hydrolyzed with 10 mL methanol/H_2_SO_4_ (90:10, *v*/*v*) or 5 mL methanol/HCl (90:10, *v*/*v*), at concentrations of 1 and 4 M (both), for 20 h at 85 °C [14]. The samples were centrifuged (5000 rpm, 10 min). The supernatants obtained as well as the supernatants of the EP were used for the analysis of total phenolic content, total flavonoid content, condensed tannins, and polyphenols by MS-UPLC-QTOf.

### 2.4. Total Phenolic, Flavonoid, and Condensed Tannins 

To determine total phenols in the samples, spectrophotometric methods were performed to measure the total phenolic content (TP) [15], total flavonoid content (TF) [16], and condensed tannin content (CT) [17]. The results were expressed as mg gallic acid equivalent for TP, mg (+)-catechin equivalent per 100 g for TF and mg (+)-catechin equivalent/kg fresh weight (+CE/kg, FW) for CT, after comparison with standard curves.

### 2.5. Betanin and Vulgaxanthines 

The content of betanins and vulga-xanthins in the lyophilized fresh mass of berrycactus was determined spectrophotometrically using the equations of Nilsson [18]: % betanins = ((α/1129) × DF × 100), % vulga − xanthins = ((γ/750) × DF × 100);
where
A = 1.095(A_538_ − A_600_) = γA_476_ − (A_538_ − α) − (α/3.1),
and
DF = dilution factor.

The percentage of betanins and vulga-xanthins was converted to g/kg fresh weight.

### 2.6. Dietary Fiber

Dietary fiber was determined according to [19], with slight modifications. The analysis was carried out in duplicate, generating two residues. The previously defatted sample was treated with the enzyme α-amilase, then incubated to 95 °C, the pH adjusted, and a protease solution added; incubated again at 60 °C, the pH adjusted again, and the enzyme amyloglucosidase added; and finally 95% ethanol added and allowed to rest for 24 h to complete precipitation. The supernatants were dried, then one residue was used for protein determination and the other for ash determination.

The % of total dietary fiber was calculated as follows: % total dietary fiber = (average R1 and R2 (mg) − protein (mg) − ash (mg) × 100

### 2.7. Antioxidant Capacity Determination 

Antioxidant capacity (AC) was determined by the ability of samples to scavenge radicals by ABTS, DPPH, ORAC, and FRAP techniques in extractable and non-extractable samples [16,20,21,22]. Antioxidant capacity was expressed as μmol Trolox equivalents/g freeze-dried extract (μmol TE/g LE), using a standard curve for Trolox.

### 2.8. Identification of Bioactive Compounds by UPLC-QTOF-MSE

Sample preparation for chromatographic analysis was as follows: 1.5 mL of the supernatants of EPs and NEPs were collected and placed in microcentrifuge tubes. The solvent was evaporated using the SpeedVac device (Thermo Scientific Mod. SAVANT SC2010A, Waltham, MA, USA) for 3 days, resuspended in methanol (200 μL) and filtered with a 0.22 μm membrane [23]. The phytochemical profile was evaluated in an Ultra-Performance Liquid Chromatograph (UPLC) coupled with a diode array detector (DAD) and a quadrupole time-of-flight (Q-ToF) mass spectrometer (MS) with electrospray ionization (ESI) (Vion IMS, Waters Co). Samples were filtered (0.45 mm) and injected directly into a BEH Acquity C18 column (2.1 × 100 mm, 1.7 mm) at 35 °C. For chromatographic separation, water with 0.1% formic acid (A) and acetonitrile (B) were used as the mobile phase at a flow rate of 0.5 mL/min. The gradient conditions were 0% B/0 min, 15% B/2.5 min, 21% B/10 min, 90% B/12 min, 95% B/13 min, 0% B/15 min, and 0% B/17 min. The absorbances were measured at 214, 280, 320, 360, 484, and 535 nm. The following commercial standards were used for quantification: Eriocitrin (flavanones), genistein (isoflavones), quercetin (flavonols), p-hydroxybenzoic acid (hydroxybenzoic acids), ferulic acid (hydroxycinnamic acids), betanin (betalains), beta-sitosterol (phytosterols), and alpha-tocopherol (tocopherols). The results were expressed in μg/g of freeze-dried fruit. The following MS conditions were used: capillary voltage, 2.0 kV; cone voltage, 40 eV; low collision energy, 6 V; high collision energy, 15–45 V; source temperature, 120 °C; cone gas flow, 50 L/h; desolvation gas, N_2_ at 450 °C and 800 L/h. Data acquisition was performed in negative ionization mode (ESI-) in a mass range of 100–1200 Da. Leucine enkephalin solution (50 pg/mL) was used for lock mass correction at 10 mL/min. Identification was performed by analyzing the exact mass of the pseudomolecular ion (mass error < 10 ppm), isotopic distribution, and fragmentation pattern.

### 2.9. In Vivo Experimental Design

The chemoprotective effect of whole berrycactus and a BCR thereof was evaluated in an animal model on the prevention of the development of preneoplastic lesions in an early cancer model of the colon. The garambullo residue was obtained according to what was reported by Rodríguez-González et al. [24]. First, the freeze-dried garambullo powder was treated using water at 1:5 ratio (wet weight/solvent, *w*/*v*) then stirred for 20 min at 21 °C. Sample was filtered and subsequently dried in a convection oven (BINDER ED115, Germany) at 60 °C for 24 h, and then ground.

Briefly the in vivo experiment was as follows: 4-week-old male Sprague Dawley rats with an initial weight of 69.6 ± 5 g were used. The rats were kept at a room temperature (21 ± 2 °C) and, a 12/12 h light/dark cycle and water and basal diet ad libitum for one week. After acclimatization, rats were randomly divided into 6 treatment groups (n = 12) (Figure 1). Negative control (C−) group: basal diet, subcutaneous saline injection; AOM/DSS (azoxymethane + dextran sodium sulfate) group: basal diet plus a subcutaneous injection of AOM (10 mg/kg body weight dissolved in 1 mL of physiological solution) once a week for 2 weeks (weeks 3 and 4), and at the same time a 7-day cycle of 2% dextran sodium sulfate (DSS) dissolved in their water ad libitum; BC (berrycactus) group: 5 g/kg body weight daily of berrycactus and basal diet; BCR (berrycactus residue) group: 5 g per kg body weight daily of berrycactus residue and basal diet; BC+AOM/DSS group: BC plus AOM/DSS and basal diet, and BCR+AOM/DSS group: BCR plus AOM/DSS and basal diet. BC and BCR were administered orally via a pellet prepared by mixing the treatment dose in the basal food normally ingested by the rat (RodentLab Chow 5001). The BC and BCR were administered to rats once daily during the experimental period (16 weeks) and the dose were chosen according to the dose previously reported as a safe dose [25].

Food and water intake and weight gain were recorded weekly. At the end of the experimental period, the animals were sacrificed with CO_2_ under the ethical method reported by Kulkarni et al. [26], and the feces of each group were collected, frozen in liquid nitrogen and stored at −70 °C to later determine the concentration of SCFA. Histological analyzes were also performed to determine the development of aberrant crypt foci (ACF) by staining with methylene blue and hematoxylin and eosin. All procedures were performed according to NOM-062-200-1999 and the project was approved by the Bioethics Committee of the Autonomous University of Querétaro (number 11522).

### 2.10. Histologic Analysis

The distal colon of the sacrificed animals was removed, and the identification and quantification of the ACF was performed by staining with methylene blue. The distal colon of the animals was removed by longitudinal section from the cecum to the anus, washed with saline solution, cut into a 1 cm × 1 cm square, and preserved in 10% buffered formalin under refrigeration. Specimens were stained with 0.5% methylene blue for 10 min, rinsed with physiologic solution to remove excess dye, and ACF were observed with an inverted light microscope (Velab, VE-BC3PLUS, MEX), expressed per cm^2^ of distal colon [27]. Multiplicity analysis was performed and the number of crypts per focus was determined and classified as 1, 2, 3, and ≥4. The architecture of the distal colon was analyzed by staining with hematoxylin and eosin: Another portion of the distal colon was harvested. It was preserved in phosphate buffer solution, fixed with 4% formaldehyde and paraffinized. After deparaffinization, they were stained with hematoxylin and eosin and analysis was performed in an optical microscope (Leica DM750 Microsystems, Switzerland) [28].

### 2.11. β-Glucoronidase Activity Determination

The β-glucoronidase enzyme activity was determined using the β-glucoronidase activity fluorometry assay kit (ab234625). Briefly, 10 mg of fecal contents were homogenized on an ice bed with a polytron and 100 μL of XLIII/β-glucoronidase buffer solution. The lysate was centrifuged at 10,000× *g* for 5 min at 4 °C, the supernatant was collected and 2 µL was added to a 96-well plate. The volume of each reaction was adjusted to 90 µL with the XLIII/β-glucoronidase buffer. The positive control was prepared with 5 µL of the reconstituted positive control and the volume was adjusted to 90 µL with the XLIII/β-glucoronidase buffer. For the substrate mixture, the substrate stock solution was diluted 10-fold with the XLIII/β-glucoronidase buffer to obtain the working solution. To start the reactions, 10 µL of the working solution was added to the positive control and samples. Fluorescence was then measured (Ex/Em = 330/450 nm, fluorescence microplate reader; Variouskan Flash Multimode Reader, Thermo Fisher)) immediately after addition of the substrate for 0–60 min at 37 °C. The β-glucoronidase activity was quantified using a standard curve of 0, 0.4, 0.8, 1.2, 1.6, and 2.0 nmol 4-methylumbelliferone (4-MU).

### 2.12. Short-Chain Fatty Acids in Feces

The production of SCFA in feces was determined using the method proposed by García-Villalba et al. [29]. In brief, 1 g of feces was mixed in 10 mL of a 9% formic acid solution and centrifuged at 10,000 rpm for 15 min. The supernatant was collected, and 10 mL of ethyl acetate was added, shaken for 2 min and centrifuged again. The organic phase was collected and transferred to an injection vial. A volume of 1 microliter of the sample was injected into the gases chromatograph (J&W Hp 88M column, length 100 m, I.D. 0.32 mm, film 0.05 µm) at an injector temperature of 250 °C. The initial column temperature was 90 °C, which increased to 150 °C at 15 °C/min, to 170 °C at 5 °C/min, and finally to 250 °C at 20 °C/min, and was maintained at this temperature for 2 min (total time 14 min). The detector was operated in electron impact ionization mode (electron energy 70 eV) and scanned the range of 30–250 *m*/*z*. The temperatures of the ion source, quadrupole, and interface were 230 °C, 150 °C, and 280 °C, respectively. Standards were used to identify the SCFAs, taking into account the retention time of the standards and the area under the curve.

### 2.13. Statistical Analysis

One–way analysis of variance (ANOVA) was performed using the statistical package SPSS v.23 and Graph prism v.7 for Mac with the post hoc Tukey test for multiple comparisons, considering significant differences at *p* ≤ 0.05. All data were reported as means ± SD of three analyzes (*n* = 3) for each chemical determination.

## 3. Results and Discussion

### 3.1. Betalains and Dietary Fiber

The results showed a content of 25.2 ± 0.9 mg/kg fresh weight (FW) of betanins (responsible for the purple coloration) and 2.9 ± 0.1 mg/kg FW of betaxanthins (responsible for the yellow pigmentation) in berrycactus. Betalains are responsible for the color of berrycactus [30]. These results are consistent with those reported by Guzmán-Maldonado et al. [4] (29.5 ± 1.9 and 2.89 ± 0.1 mg/kg, respectively). Spectrophotometric analysis of betalains was also performed in acid hydrolysis, but they were not found, most likely because they are easily soluble pigments that were completely extracted in the extractable fraction or destroyed by the acidic environment. On the other hand, the dietary fiber content of BC was 29.8 ± 1.2 g/100 g FW, which corresponds to the value of 34.9 ± 1.61 g/kg FW of fruit reported by Guzmán-Maldonado et al. [4]. The fiber content of berrycactus was higher than that of *Hylocereus polyrhizus* (“pitahaya”) with 15.9 g/kg fruit and similar to that of prickly pear (*Opuntia ficus-indica*) with 32–36 g/kg fruit [31,32]. 

### 3.2. Phenolics in Extractable and Non-Extractable Polyphenols 

The amount of extractable TP and TF was 958.7 ± 26 and 397.1 ± 13, respectively, and were lower to those compared for NEP of BC (Table 1); for example, an increase up to 116% and up to 360% of TP and TF, respectively, both in H_2_SO_4_ and HCl hydrolysis was observed. A slight increment of CT in 1 M H_2_SO_4_ extraction was observed (Table 1). It is clearly that TP and TF remain bound to the polysaccharides of BC matrix through hydrogen bonds, phenol–fiber interactions, van der Waals, covalent bonds, and/or ether bonds [33]. 

Levels of extractable TP reported here were similar to those reported by Guzman-Maldonado et al. [4] and Herrera-Hernández et al. [12], when results are compared on dry weight basis. On the other hand, non-extractable TP content of HCl hydrolysis of BC (2070.1 ± 14.1 mg GAE/100 g) was higher than previously reported [23], in hydrolyzable polyphenols of fruits such as apple (1062.7 ± 25.4 mg GAE/100 g), banana (1415.1 ± 206.5 mg GAE/100 g), orange (1124.7 ± 61.3 mg GAE/100 g), and pear (1427.1 ± 38.7 mg GAE/100 g). These results could be due to the fact that berrycactus has a higher fiber content than these fruits.

### 3.3. Polyphenol Profile of EP and NEP by UPLC-QToF MS

The identification of 33 polyphenolic compounds corresponding to EP and NEP is shown in Table 2. These compounds include nine flavonoids, ten hydroxybenzoic acids, nine hydroxycinnamic acids, and four betalains. The present work reports for the first time the following EP and NEP in berrycactus: kaempferol hexoside, dihydroxybenzoic acid isomer I, methyl gallic acid, dihydroxybenzoic acid hexoside, dihydroxybenzoic acid isomer II, gallic acid, hydroxybenzoic acid isomer III, coumaroyl hexoside, and caffeic acid.

The phenolic compounds that were present in greater amounts in EP compared to NEP were glycosylated phenols, which are more soluble in aqueous extractions, and, as expected, betalains. Betanidin β-hexoside is the most important phytochemical in EP together with quercetin rhamnosyl-rhamnosyl-hexoside (157.64 ± 3.01 and 150.48 ± 0.73 µg/g, respectively), while in NEP the major compounds in the extraction with 1 M and 4 M HCl were hydroxybenzoic acid isomer II followed by quercetin rhamnosyl-rhamnosyl-hexoside (76.01 ± 0.72 and 40.74 ± 0.04, µg/g, respectively). Hydrolysis with 1 M HCl extracted higher amounts of most of the identified compounds compared to extraction with 4 M HCl. Many of the identified EP decreased in concentration or were undetected after treatment with acid hydrolysis, with the exception of hydroxybenzoic acid isomer II, which increased after hydrolysis. 

Some phenolic compounds were identified as part of NEP only upon hydrolysis, namely isorhamnetin, quercetin, dihydroxybenzoic acid isomer I, methyl gallic acid, didydroxybenzoic acid isomer II, gallic acid, and caffeic acid (Table 2). As mentioned above, this is most likely since these compounds were bound to the fruit matrix. The results of the identification of compounds in the EP are consistent with what was reported by Reynoso-Camacho et al. [34] in the acid hydrolysis of some berries (strawberries, raspberries, blueberries, and blackberries), such as kaempferol rutinoside, myricetin, quercetin, kaempferol hexoside, coumaric acid, and some hydroxybenzoic acids, among others, although they differ in quantity.

The different types of hydroxybenzoic acids found in BC NEP were higher than those reported by Reynoso-Camacho et al. (2021), which ranged from 1.34 ± 0.01 to 5.14 ± 0.04 µg/g in blueberries and 9.823 ± 0.157 µg/g in blackberries, while these compounds ranged from 2.50 ± 0.08 to 76.01 ± 0.72 µg/g in the 1 M hydrolysis of BC and from 0.98 ± 0.02 to 31.99 ± 0.25 µg/g in the 4 M hydrolysis. Fonseca et al. [35] reported gallic acid and ferulic acid (10.4 and 2.6 µg/g, respectively) in purple passion fruit, and El-Hawary et al. [36] reported 20.24 µg/g of quercetin rutinoside in prickly pear (*Opuntia ficus*-*indica*), concentrations lower than those reported here. Both passion fruit and prickly pear fruits are derived from cacti, and are therefore related to berrycactus. The betalains, identified as betanidin, indicaxanthin, betanin, and isobetanin (Table 2), were also found in the acetone–water extract of berrycactus by Sánchez-Recillas et al. [5], and in a methanol–water extract with ultrasonication by Montiel-Sánchez et al. [37]. No betalains were identified in the acid hydrolysis, which is consistent with the spectrophotometric results.

There are no previous reports of TF or CT measured spectrophotometrically and NEP from BC. These results are relevant to the nutraceutical properties of BC.

### 3.4. Antioxidant Capacity of EP and NEP

Antioxidant results are summarized in Table 3. The DPPH assay simulates the antioxidant reaction with the peroxyl radical of a biological system through transient nitrogens [38], TEAC and ORAC are used to evaluate the ability of components to transfer electrons and hydrogen atoms, respectively, while FRAP is a typical method based on the single electron transfer reaction (Fe^+3^ to Fe^+2^) [39,40]. The ORAC assay measures the radical chain reaction ability of antioxidants by monitoring the inhibition of peroxyl radical oxidation, which is characterized as a free radical prevalent in lipid oxidation in biological systems under physiological conditions, and therefore ORAC values are considered biologically relevant [41]. 

There are no previous reports on the antioxidant capacity of NEP from BC. In the DPPH assay, NEP obtained by acid hydrolysis with HCl (1 M) showed the highest antioxidant capacity (452 ± 13 μmol TE/g), greater than the extractable polyphenols (40.0 ± 1.5 μmol TE/g). The DPPH radical is better solubilized in alcoholic media than in aqueous media [42], which would explain the increase in antioxidant capacity upon acid hydrolysis associated with the release of components of the BC matrix that contribute to antioxidant capacity.

The TEAC value of the EP of BC was 100.8 ± 3.6 μmol Trolox equivalent/g, higher than that of fruits such as blackberries, pomegranates, and strawberries (47, 43.5 and 34.9 μmol Trolox equivalent/g, respectively [43]. The TEAC values of BC NEP ranged from 525.9 ± 51 to 1560 ± 66 μmol Trolox equivalent/g and were thus higher than the TEAC value of EP. The higher TEAC value was observed when hydrolyzed with 1 M HCl. The TEAC values of NEP from BC are consistently higher than the values of EP, which is consistent with previous reports by Tow et al. [44] on apple wastes in which NEP had significantly higher radical scavenging capacity than EP (68% higher).

The ORAC value of EP was 140.5 ± 5.5 μmol Trolox equivalent/g, similar to 113.2 μmol Trolox equivalent/g previously reported for BC by Herrera-Hernández et al. [12]. Similar values are reported for blueberries with 101.7 μmol Trolox equivalent/g and lower values for fruits such as red grapes, tangerines, and melons (26.05, 37.1 ± 1.2, and 16.4 ± 0.9 μmol Trolox equivalent/g, respectively) [23,45]. The highest value of NEP was obtained in hydrolysis with 1 M HCl acid at 1620.8 ± 63 μmol Trolox equivalent/g. This result is higher than that reported for cranberry with an ORAC NEP value of 1258.53 μmol Trolox equivalent/g [46]. 

The FRAP values for NEP extracted with 1 M HCl were 11.3 times higher than the FRAP values for EP. The highest result in this study was 1199 ± 38 μmol TE/g DM, which is higher than the FRAP values in NEP from pear of 740 μmol FeSO_4_/Kg FW reported by Liu et al. [47]. The FRAP value for EPs from BC was 105.7 ± 3.3 μmol TE/g, showing a greater antioxidant capacity than other fruits such as strawberry and cherry 32.9 ± 0.30 and 45.8 ± 0.62 μmol TE/g, respectively [48]. 

The release of phenolic compounds by the acid hydrolysis performed in this study affects the increase in antioxidant capacity by the different methods evaluated. The compounds contained and released in the NEP have an electron and hydrogen transfer capacity as well as a ferrous ion reduction capacity, which gives them an attractive property that can be beneficial in biological models.

### 3.5. In Vivo Model

The water and feed intake as well as the weight gain of the animals were recorded weekly. They were similar in the six groups over time, with no statistically significant difference between the groups. No signs indicative of necrosis, diarrhea, vomiting, or significant weight loss were observed in the animals during the experimental period.

### 3.6. Quantification of ACF in Distal Colonic Tissue

The identification of ACF was based on the literature as enlarged pericryptal, oval crypts, thickened epithelium, darker staining than normal crypts, more than one focus per crypt, and protrusion from the focal plane [49,50], and is shown in Figure 2A, which confirms that the dose of both treatments (5 g/kg weight) does not contribute to the development of early lesions as it does not induce the development of ACF. On the other hand, significantly fewer ACF were found in the BC+AOM/DSS and BCR+AOM/DSS groups in the colon section analyzed (5.9 ± 1.2 and 5.2 ± 1.6 ACF, respectively) than in the AOM/DSS group (9.7 ± 1.5 AFC). In addition, the multiplicity of ACF was analyzed (Figure 2B), which refers to the number of crypts per focus and indicates that the greater the multiplicity (more than four crypts per focus), the greater the likelihood of adenoma and adenocarcinoma formation [51]. BC+AOM/DSS and BCR+AOM/DSS showed a higher number of ACFs with low multiplicity (2C), while AOM/DSS induced ACFs with higher multiplicity (3C and 4C), indicating a higher potential for development into malignant lesions in the colon. In addition, significant differences were found between the BC+AOM/DSS (0.37 ± 0.42 AFC) and BCR+AOM/DSS (0.08 ± 0.30) groups at the highest multiplicity (four crypts per focus) compared to the AOM/DSS group. Although the multiplicity for three crypts/focus was not significant, it can be observed that the AOM/DSS group has a greater number of three crypt per foci than the treatment and chemically induced groups.

Previously, other berries such as blueberry and pomegranate had shown a chemoprotective effect against the development of ACF in a cancer model treated with AOM for 16 weeks. The authors found a total number of ACF per colon of 11.33 ± 2.85 and 15.67 ± 1.86 for blueberry and pomegranate juice, respectively, versus 171.67 ± 5.60 in the AOM group [52]. Similarly, Arango-Varela et al. [53] reported a significant decrease in the development of ACF in mice treated with AOM compared to the control group when treated for 8 weeks with Andean berries, an underutilized endemic fruit from Colombia that contains dihydroxybenzoic acid and gallic acid as major constituents, the same constituents identified in the characterization of BC in this study.

The results indicate that consumption of BC and BCR two weeks prior to exposure to AOM/DSS prevents the development and proliferation of ACF, suggesting a chemopreventive effect in the early stages of colon cancer. Our results show that the BC matrix contains dietary fiber and phytochemicals that are thought to act at the molecular level to prevent the occurrence of ACF. Therefore, it could be hypothesized that undigested compounds in the upper part of the gastrointestinal tract reach the colon and prevent the damage caused by the substances used as triggers and promoters of cancer, or even enhance the molecular mechanisms of cellular defense that prevent the development of primary colon cancer lesions.

Histologic analysis of the distal colon was also performed with hematoxylin and eosin staining. Figure 3A shows the frequency of inflammation and premalignant damage in the colon between the groups, and Figure 3B shows representative images of the distal colon tissue of the groups. In the control group, the preserved architecture of the crypts and the preservation of the epithelium can be seen. 

The image of the AOM/DSS group shows a distortion of the architecture of the crypts, manifested by shortening and incipient branching, as well as an increase in proliferation at the base, loss and fusion of crypts, dysplasia of the goblet cells (hyperchromatic masses) and infiltration of lymphocytic cells in the lamina propria. AOM is a carcinogenic agent that is metabolized in the liver to cytochrome P450 (CYP2E1 isoform) and produces a highly electrophilic alkylating agent that induces O6-methyl-guanine adducts in DNA that generate G/A transition mutations. These mutations alter the Kras and CTNNB1 genes and contribute to the development of preneoplastic lesions through the uncontrolled proliferation of initiated cells [54]. On the other hand, the BC+AOM/DSS and BCR+AOM/DSS treatments were able to preserve the architecture of the colon, mucosa, and submucosa, and reduce the dysplasia and lymphocytic infiltration that were clearly observed in the AOM/DSS group. AOM/DSS is a trigger for aberrant crypt foci and inflammation [55], which is reflected in the results of the present study, in which treatment with AOM/DSS induced 35.6% inflammation and 33.0% FCA and altered the architecture of the colonic tissues of the rats. Co-administration of BC and BCR reduced damage by 72.5 and 39.7%, respectively, compared to the AOM/DSS group, with partial preservation of tissue architecture, but no atrophy was observed. Several studies, both in vivo and in vitro, suggest that polyphenols from various dietary sources are capable of delaying cancer development and progression by reducing cell proliferation, inactivating carcinogens, inducing cell cycle arrest and apoptosis, and promoting immunomodulation. These compounds have been shown to inhibit the action of NF-κB, protein kinase C, and AP-1 activating protein, and induce the endogenous antioxidant system through the induction of Nrf2 [56]. 

According to the characterization reported in this study, BC contains bioactive compounds that could induce protective molecular effects against carcinogenic damage in the colon. This is the case of gallic acid, which showed anticancer activity in Caco-2 cells by MTT assay, exhibiting significant cytotoxic activity (45%) compared to the control [57]. Secme et al. [58] reported the antiproliferative, apoptotic, and anti-invasive effects of caffeic acid in HCT116 colon cancer cells by regulation of the expression of genes related to apoptosis, cell cycle, and invasion, as well as microRNAs, and reduced oxidative stress and GST enzyme activity. In addition, quercetin, a flavonoid found in glucosylated and non-glucosylated form in BC by UPLC-MS, was extensively studied, demonstrating inhibition of cell proliferation by inducing apoptosis and/or cell cycle arrest through the antioxidant and anti-inflammatory effects. Recent studies have reported that the cancer-preventive effects of quercetin can be attributed to its antioxidant activity, inhibition of carcinogenic enzymes, regulation of intracellular signal transduction pathways, interaction of quercetin with receptors, etc. For example, Tezerji et al. [59] reported that quercetin ingestion reduced cytologic changes in colon cancer cells in a rat model by decreasing the expression of beta-catenin and Bcl-2 proteins and increasing the expression of caspase 3 compared to the control. Betalains purified from *Beta vulgaris* (beetroot) significantly reduce COX-2 and IL-8 mRNA expression levels in CaCo-2 cells and have a cytotoxic effect (IC_50_ at 25 μg/mL), which could be proposed as dietary ingredients to reduce and limit the initiation and development of colorectal cancer. As mentioned above, the DPPH technique measures the neutralization reactions of nitrogen-derived free radicals [38]. The compounds found in the characterization of BC showed antioxidant capacity using this technique. This suggests that they could reduce the toxicity of the carcinogen AOM, as it generates a nitrogen radical during its metabolism in the liver. AOM is metabolized in the liver to methylazoxymethanol (MAM), glucuronidated for excretion, and cleaved from its glucuronide in the colon by the enzyme b-glucuronidase, allowing it to exert its carcinogenic effects [60]. The phenolic compounds contained in BC could reduce the activation of AOM by neutralizing MAM once it is released in the colon. This is the first report on the effect of BC in the prevention of preneoplastic lesions in the colon.

### 3.7. β-Glucoronidase Activity

The results showed that the daily administration of BC (5 g/kg body weight) and BCR were sufficient to reduce bacterial β-glucuronidase activity in feces (49.3 ± 4.0 and 45.5 ± 3.7 pmol/min/mL sample, respectively) compared to the AOM group (85.0 ± 2.2 pmol/min/mL sample) (Figure 4). 

The activity of the enzyme β-glucoronidase is important in the process of colon carcinogenesis, as it is able to hydrolyze glucuronide conjugates, releasing and reactivating carcinogenic metabolites in the intestine [61]. After injection, AOM is biotransformed by phase I enzymes to form methylazoxymethanol (MAM), which is conjugated with glucuronic acid by UDP-glucoronosyltransferase (UGT) and excreted in the feces. In the intestine, the conjugate is cleaved by the enzyme β-glucuronidase, which reactivates the carcinogenic potential of AOM [60]. It has already been shown that the high activity of the enzyme β-glucuronidase in rats contributes to the increase in the number of ACF [62]. Therefore, the reduction in β-glucuronidase activity in the groups administered BC and BCR and induced with AOM could contribute to the chemoprotection mechanism reflected in the prevention of ACF development in both groups in the animal model used. The reduced β-glucoronidase activity is probably due to the reduction of the pathogenic bacterial population that produces a higher amount of β-glucoronidase in the colon, such as *Escherichia coli, Clostridium perfringens*, and *Bacteroides* species [63]. 

### 3.8. Production of Short-Chain Fatty Acids

The production of short-chain fatty acids (SCFA) was monitored in vivo at the beginning, middle, and end of the experimental phase, and is shown in Figure 5. The amount of SCFA is similar at the beginning of the experimental period, then increases over time, with the highest concentration of acetic acid, followed by propionate, and the lowest concentration of butyrate in all groups at 16 weeks. After 8 weeks, acetate and butyrate increased in the BC and BCR groups (12.08 ± 1.92 and 11.79 ± 0.67 µM of acetate, 7.49 ± 0.84 and 7.41 ± 0.27 µM of butyrate, respectively), while the AOM group had a higher acetate content after 16 weeks (17.30 ± 0.51 µM), and the C- and BCR groups had a higher butyrate content (7.09 ± 0.90 and 7.88 ± 0.74 µM, respectively). Groups AOM, BC, BCB+AOM/DSS, and BCR+AOM/DSS had a lower concentration of this acid at the end of the experiment (5.74 ± 0.58, 5.33 ± 0.52, 5.81 ± 0.07, and 5.81 ± 0.81 µM, respectively). 

Among all the produced SCFAs, butyrate is rapidly oxidized to generate fuel after uptake into enterocytes in differentiated intestinal epithelial cells and contribute to the proliferation of healthy mucosa. However, there is a discrepancy in the function of butyrate in neoplastic and undifferentiated cells, as it is not utilized as an energy source since they preferentially use glucose, and its accumulation in the cytoplasm leads to modification of histones (inhibition of HDAC) and cell cycle arrest. This effect is known as the “butyrate paradox” [64,65]. This would explain why the BC+AOM/DSS and BCR+AOM/DSS groups have similar butyrate concentrations at the end of the experimental period. On the one hand, the AOM/DSS group has an accumulation of butyrate produced by bacterial metabolism without being excreted in the feces. On the other hand, the BC+AOM/DSS and BCR+AOM/DSS groups utilize the butyrate produced in molecular processes that result in stopping proliferation and protecting the colonocytes, which is reflected in a significant decrease in the development of ACF in this study. Sze et al. [66] reported that fecal SCFA concentrations do not differ between healthy human colons and those with colon adenomas or carcinomas. Furthermore, fermentation of dietary fiber is accompanied by chemical changes in phenolic compounds, producing hydroxylated SFCAs such as 3,4 diOH-phenylacetic acid, 3-phenylpropionic acid, hydroxybenzoic acid and others [67]. Other authors hypothesize that they play a role in triggering cell death, differentiation, and/or growth inhibition of transformed colonocytes by regulating the level of messenger RNA of genes involved in these processes [68,69]. In the present study, these SFCAs were not identified, but it is hypothesized that they are produced and that they may induce protective molecular mechanisms in the induced groups that consumed BC and BCR. The identification and characterization of hydroxylated SFCAs in feces is a perspective that will be analyzed later in this project.

## 4. Conclusions

The present work reports for the first time the following EPs and NEPs in berrycactus as kaempferol hexoside, dihydroxybenzoic acid isomer I, methyl gallic acid, dihydroxybenzoic acid hexoside, dihydroxybenzoic acid isomer II, gallic acid, hydroxybenzoic acid isomer III, coumaroyl hexoside, and caffeic acid. HCl acid hydrolysis with 1 M was able to extract the highest content of total phenols, total flavonoids, and total tannins from berrycactus, while no betalains were identified with acid hydrolysis. NEP showed higher antioxidant capacity compared to EP as measured by DPPH, TEAC, FRAP, and ORAC methods. During the in vivo experiment, daily consumption of berrycactus or berrycactus residue for 16 weeks prevented the development of early-stage colon cancer in rats induced by AOM/DSS. This was reflected in a lower number of ACF, lower multiplicity, preservation of mucosal architecture and submucosa of the colon, and reduction of dysplasia and lymphocytic infiltration compared to the AOM/DSS group. The results could be explained by the decrease in the enzymatic activity of β-glucoronidase, while the formation of SCFA did not differ among groups. The results suggest that the consumption of berrycactus is an attractive option for the prevention of colon cancer, providing added value to its consumption.

## Figures and Tables

**Figure 1 antioxidants-13-01112-f001:**
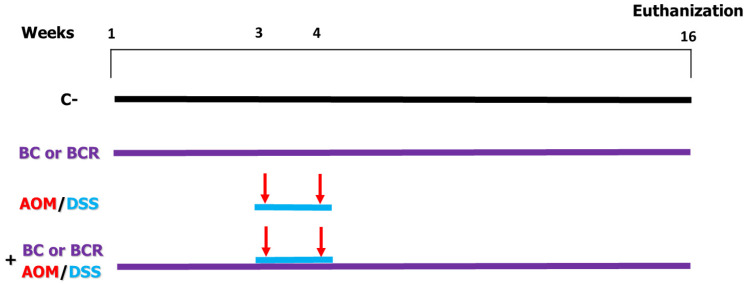
In vivo experimental design. Evaluation of the chemoprotective effect of BC or BCR in AOM/DSS-induced colorectal cancer Sprague Dawley rats. C−: negative control (saline solution 0.9% *v*/*v* injection); BC: berrycactus (5 g/kg BW); BCR: berrycactus residue (5 g/kg BW); AOM: azoxymethane (10 mg/kg BW, via intraperitoneal); DSS: dextran sulfate sodium (2%, via oral, daily for 7 days in drinking water ad libitum). BC and BCR were administered orally mixed with diet pellets. All mice consumed basal diet (Rodent Lab diet) ad libitum.

**Figure 2 antioxidants-13-01112-f002:**
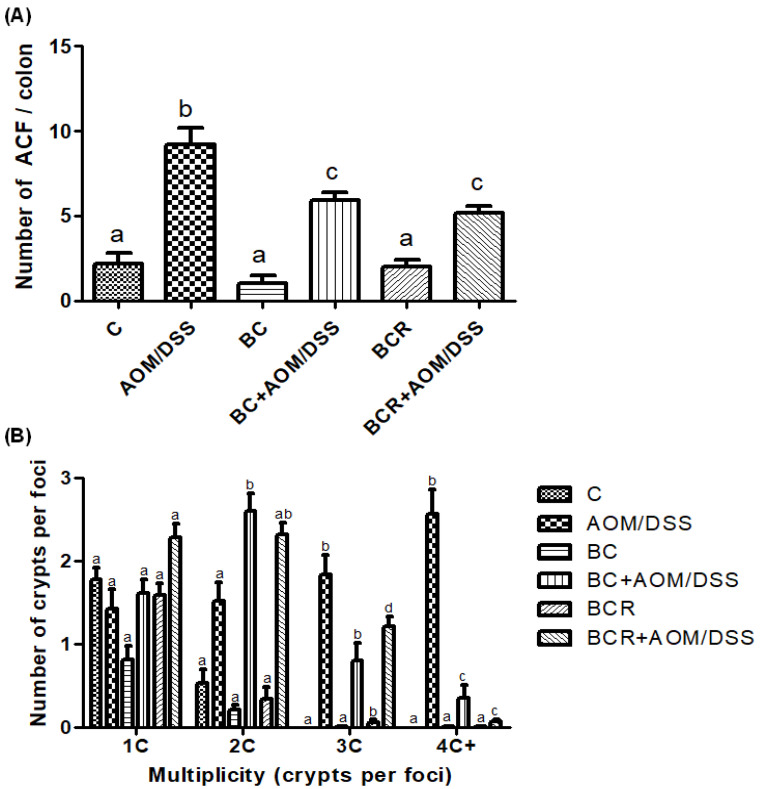
Normal and aberrant crypt foci of the distal colon with methylene blue staining. (**A**) Number of ACF in the distal colon (1 cm^2^). (**B**) Multiplicity counted by number of crypts per focus in the distal colon. Results are expressed as mean of n = 12 rats ± SD. Different letters per column indicate a significant difference *p* = 0.05, Tukey test. C-: negative; AOM: azoxymethane; DSS: Dextran sulfate sodium; BC: berrycactus; BCR: berrycactus residue.

**Figure 3 antioxidants-13-01112-f003:**
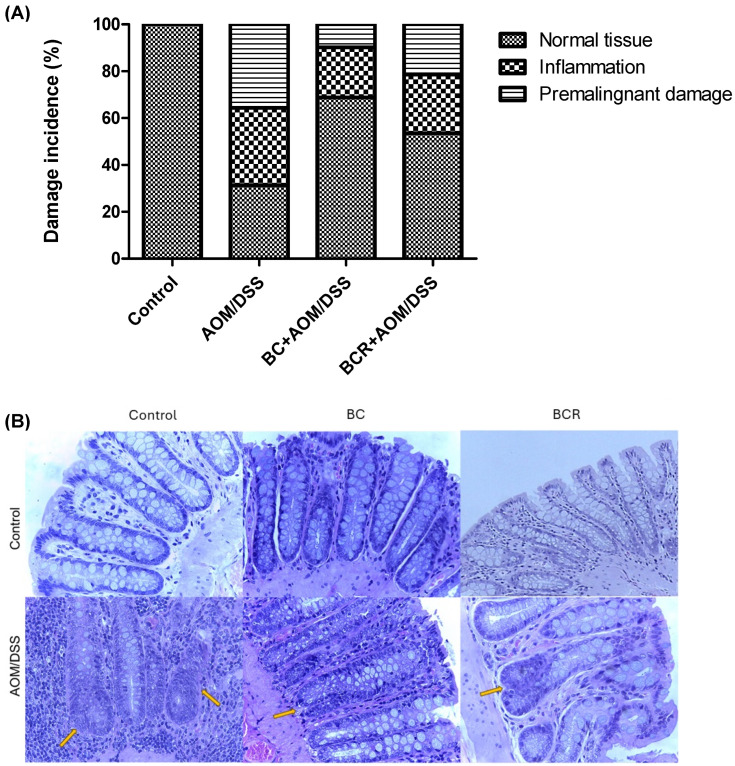
Histopathologic analysis of preneoplastic colonic lesions caused by AOM/DSS and treated with G and R. (**A**) Incidence of inflammation and premalignant damage in the colon; (**B**) Hematoxylin–eosin-stained histopathologic analyzes of colon lesions (40×). Orange arrows indicate aberrant crypt foci. AOM: azoxymethane; DSS: dextran sulfate sodium; BC: berrycactus; BCR: berrycactus residue.

**Figure 4 antioxidants-13-01112-f004:**
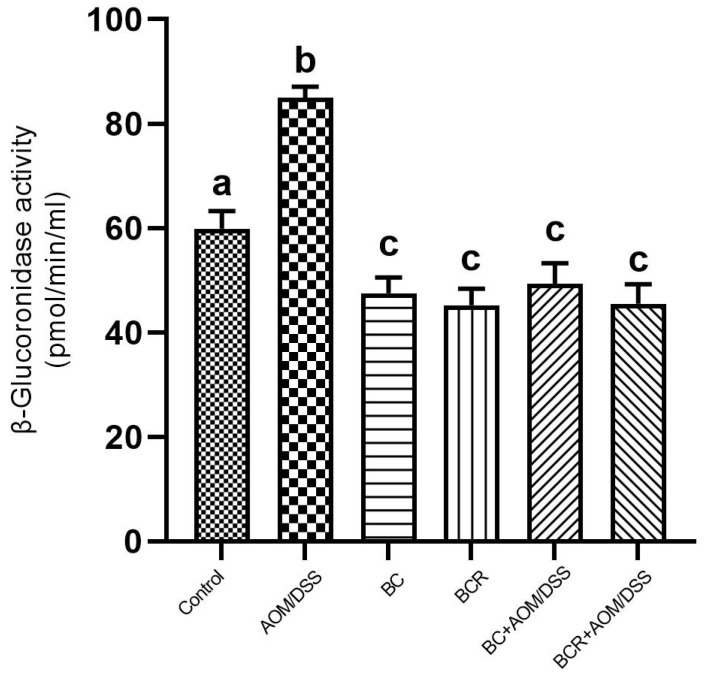
β-glucoronidase activity in experimental groups. Results are expressed as mean of two independent experiments by duplicate ± SD. Different letters per column indicate a significant difference *p* = 0.05, Tukey test. AOM: azoxymethane; DSS: dextran sulfate sodium; BC: berrycactus; BCR: berrycactus residue.

**Figure 5 antioxidants-13-01112-f005:**
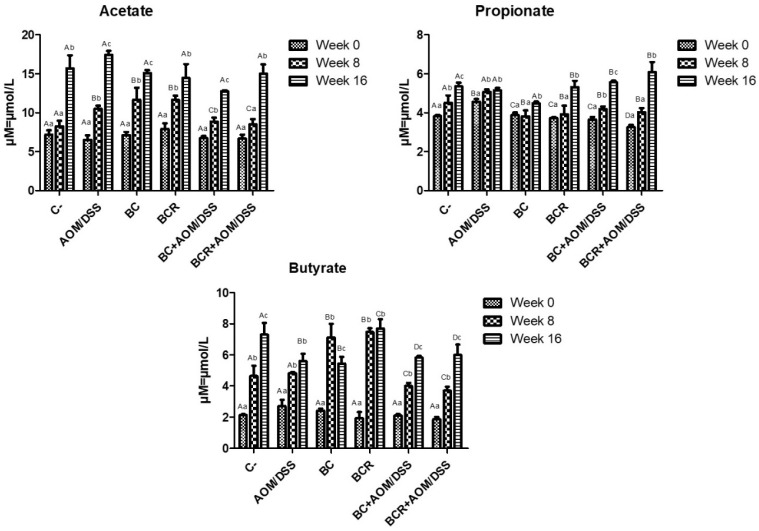
Short-chain fatty acids production in feces of experimental groups. Results are expressed as mean of two independent experiments by duplicate ± SD. Different uppercase letters indicate a significant difference between experimental groups, and different lowercase letters indicate a significant difference between times (*p* = 0.05), Tukey test. C−: negative; AOM: azoxymethane; DSS: dextran sulfate sodium; BC: garambullo; BCR: berrycactus residue.

**Table 1 antioxidants-13-01112-t001:** Extractable and non-extractable total phenol (mg GAE/100 g DW), total flavonoids (mg CE/100 g DW), and condensed tannins (mg CE/100 g) of berrycactus.

		Extractable Polyphenols	Non-Extractable Polyphenols			
			H_2_SO_4_		HCl	
			1 M	4 M	1 M	4 M
TP	(mg GAE/100 g)	958.7 ± 26 ^a^	1804.8 ± 3.7 ^b^	1447.7 ± 3.7 ^c^	2070.1 ± 14.1 ^d^	2008.1 ± 25.4 ^e^
TF	(mg CE/100 g)	397.1 ± 13 ^a^	802.2 ± 2.2 ^b^	1339.6 ± 85 ^c^	1431.0 ± 26 ^d^	1114.3 ± 11 ^d^
CT	(mg CE/100 g)	120.9 ± 4 ^a^	163.2 ± 3.7 ^b^	103.1 ± 3.7 ^c^	114.3 ± 11 ^d^	98.7 ± 3.5 ^c^

Different experiments. Different letters mean significant difference (ANOVA, post hoc Tukey test, *p* < 0.05) between treatments in the same analysis technique. GAE: Gallic acid equivalents, CE: Catechin equivalents.

**Table 2 antioxidants-13-01112-t002:** Quantification of bioactive compounds in EPs and NEPs of berrycactus by UPLC-QToF MS.

Family	Component Name	Retention Time (min)	Molecular Formula	Observed *m*/*z*	Adducts	Fragments	Extractable Polyphenols (EPs)	Non-Extractable Polyphenols (NEPs)	
								HCl	
								1 M	4 M
Flavonols	Kaempferol rutinoside	3.79	C27H30O15	593.1517	[M-H]^−^	284.02482, 151.05667	15.14 ± 3.10 ^d^	3.84 ± 0.16 ^c^	4.27 ± 0.15 ^b^
	Myricetin *	3.99	C15H10O8	317.0315	[M-H]^−^	179.03553, 151.04052	ND	ND	ND
	Quercetin rhamnosyl- rhamnosyl-hexoside	4.44	C33H40O20	755.2054	[M-H]^−^	609.14759, 300.02792, 151.00426	150.48 ± 0.73 ^i^	40.74 ± 0.04 ^e^	20.21 ± 0.15 ^d^
	Kaempferol rhamnosyl- hexoside-rhamnoside	5.14	C33H40O19	739.2113	[M-H]^−^	447.09479, 431.19371, 284.03324, 151.06481	12.02 ± 0.00 ^d^	2.53 ± 0.03 ^c^	0.89 ± 0.05 ^a^
	Quercetin rutinoside *	5.35	C27H30O16	609.1479	[M-H]^−^	300.02817, 151.00450	44.82 ± 0.06 ^h^	5.24 ± 7.41 ^c^	5.72 ± 0.13 ^b^
	Kaempferol dihexoside	5.61	C27H30O16	609.1482	[M-H]^−^	447.09449, 284.03222, 151.09310	9.13 ± 0.86 ^c^	4.58 ± 0.15 ^c^	7.80 ± 0.12 ^c^
	Kaempferol hexoside	6.22	C21H20O11	447.0954	[M-H]^−^	285.04134, 151.07177	8.66 ± 0.01 ^c^	1.32 ± 0.01 ^b^	1.21 ± 0.0 ^a^
	(Iso)-rhamnetin hexoside	7.50	C22H22O12	477.1062	[M-H]^−^	284.97552, 151.00421	4.74 ± 0.00 ^b^	2.45 ± 0.00 ^c^	1.02 ± 0.0 ^a^
	Isorhamnetin *	9.99	C16H12O7	315.0504	[M-H]^−^	315.05231, 300.02842, 285.04171, 151.00472	ND	0.34 ± 0.00 ^a^	0.24 ± 0.02 ^a^
	Quercetin *	11.02	C15H10O7	301.0365	[M-H]^−^	178.99924, 151.00423, 107.01432	ND	2.62 ± 0.02 ^c^	4.46 ± 0.18 ^b^
Hydroxybenzoic acids	Dihydroxybenzoic acid isomer I	1.84	C7H6O4	153.0187	[M-H]^−^	137.02375, 109.02909	ND	2.50 ± 0.08 ^c^	1.16 ±0.01 ^a^
	Methyl gallic acid	2.23	C8H8O4	183.0290	[M-H]^−^	169.04992	ND	2.85 ± 0.20 ^c^	1.62 ± 0.04 ^a^
	Hydroxybenzoic acid isomer I	2.71	C7H6O3	137.0243	[M-H]^−^	109.02171	7.78 ± 0.13 ^b^	ND	ND
	Dihydroxybenzoic acid hexoside	2.82	C13H16O9	315.0716	[M-H]^−^	153.01914, 137.02446, 109.02949	31.19 ± 2.42 ^g^	3.09 ± 0.07 ^c^	1.77 ± 0.04 ^a^
	Dihydroxybenzoic acid isomer II *	2.84	C7H6O4	153.0194	[M-H]^−^	137.03907, 109.02967	ND	3.36 ± 0.02 ^c^	1.47 ± 0.08 ^a^
	Hydroxybenzoic acid hexoside	3.01	C13H16O8	299.0773	[M-H]^−^	137.02452	21.28 ± 0.07 ^f^	3.38 ± 0.08 ^c^	0.98 ± 0.02 ^a^
	Gallic acid *	3.07	C7H6O5	169.0144	[M-H]^−^	125.0469	ND	3.02 ± 0.02 ^c^	8.57 ± 019 ^c^
	Hydroxybenzoic acid isomer II *	3.22	C7H6O3	137.0247	[M-H]^−^	109.02969	7.09 ± 0.42 ^b^	76.01 ± 0.72 ^f^	31.99 ±0.25 ^c^
	Hydroxybenzoic acid isomer III	4.19	C7H6O3	137.0251	[M-H]^−^	109.02996	5.70 ± 0.34 ^b^	ND	ND
	Vanillic acid *	4.69	C8H8O4	167.0357	[M-H]^−^	139.04067, 109.02987	4.35 ± 0.04 ^b^	9.91 ± 0.02 ^d^	7.04 ± 0.01 ^c^
Hydroxycinnamic acids	Caffeic acid hexoside	3.23	C15H18O9	341.0878	[M-H]^−^	179.03506, 135.04525	16.06 ± 0.27 ^e^	5.01 ± 1.73 ^c^	5.37 ± 0.06 ^b^
	Coumaroyl hexoside	3.40	C15H18O8	325.0933	[M-H]^−^	163.04038, 119.05054	4.21 ± 0.12 ^b^	0.92 ± 0.12 ^a^	1.12 ± 0.10 ^a^
	Ferulic acid hexoside	3.66	C16H20O9	355.1037	[M-H]^−^	193.05082, 178.02736, 134.03761	22.53 ± 1.32 ^f^	4.6 ± 0.05 ^c^	ND
	Coumaroylquinic acid	4.61	C16H18O8	337.0939	[M-H]^−^	191.05682, 163.05624	5.79 ± 1.25 ^b^	ND	ND
	Caffeic acid *	4.80	C9H8O4	179.0357	[M-H]^−^	109.02989	ND	2.13 ± 0.02 ^c^	0.31 ± 0.44 ^a^
	p-Coumaric acid *	5.11	C9H8O3	163.0408	[M-H]^−^	119.05085	8.75 ± 0.40 ^c^	1.62 ± 0.00 ^b^	1.00 ± 0.01 ^a^
	Ferulic acid *	5.19	C10H10O4	193.0515	[M-H]^−^	178.02806, 134.03809	3.27 ± 0.62 ^b^	0.64 ± 0.00 ^b^	1.12 ± 0.05 ^a^
	Sinapic acid hexoside	5.23	C17H22O10	385.1153	[M-H]^−^	223.09660, 209.08182	24.30 ± 0.11 ^f^	2.48 ± 0.05 ^c^	1.94 ± 0.07 ^a^
	Coumaroyl malic acid	5.43	C13H12O7	279.0519	[M-H]^−^	163.04082, 119.05093	4.55 ± 0.03 ^b^	2.12 ± 0.24 ^c^	ND
Betalains	Betanidin	2.80	C18H16N2O8	389.0965	[M+H]^+^	345.10744	7 ± 0.11 ^b^	ND	ND
	Proline-betaxanthin (indicaxanthin)	2.72	C14H16N2O6	309.1068	[M+H]^+^	389.09606	35.71 ± 0.18 ^g^	ND	ND
	Betanidin β-hexoside (betanin) *	2.66	C24H26N2O13	551.1486	[M+H]^+^	389.09643	157.64 ± 3.01 ^i^	ND	ND
	Isobetanidin β-hexoside (isobetanin)	3.67	C24H26N2O13	551.1481	[M+H]^+^	265.09656	1.98 ± 0.29 ^a^	ND	ND

* Identification confirmed with commercial standards. Data are shown as mean ± standard deviation of three replicates. Results are expressed as μg/g. Different letters indicate significant (*p* < 0.05) differences between samples. ND, non-detected.

**Table 3 antioxidants-13-01112-t003:** Antioxidant capacity (DPPH, FRAP, TEAC, and ORAC) of extractable and non-extractable polyphenols of berrycactus.

		Non-Extractable Polyphenols (μmol TE/g GL)			
	Extractable Polyphenols (μmol TE/g GL)	H_2_SO_4_		HCl	
		1 M	4 M	1 M	4 M
DPPH	40.0 ± 1.5 ^a^	376.6 ± 9 ^b^	394.2 ± 12.2 ^b^	452 ± 13 ^c^	357.2 ± 10.6 ^b^
FRAP	105.7 ± 3.3 ^a^	614.7 ± 20 ^b^	841.4 ± 30 ^c^	1199 ± 38 ^d^	1159 ± 22 ^d^
TEAC	100.8 ± 3.6 ^a^	1445.8 ± 34 ^b^	1037.9 ± 44 ^c^	1560 ± 66 ^d^	525.9 ± 51 ^e^
ORAC	140.5 ± 5.5 ^a^	509.3 ± 22 ^b^	302.7 ± 11 ^c^	1620.8 ± 63 ^d^	1464 ± 56 ^e^

Determinations ± one standard deviation. Different letters mean significant difference (ANOVA, post hoc Tukey test, *p* < 0.05) between treatments in the same analysis technique.

## Data Availability

Data is contained within the article.

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
