# Peer review of "Characterization of Extractable and Non-Extractable Phenols and Betalains in Berrycactus (Myrtillocactus geometrizans) and Its Chemoprotective Effect in Early Stage of Colon Cancer In Vivo"

_antioxidants, 2024, doi:10.3390/antiox13091112_

Round 1

Reviewer 1 Report

Some compounds have been detected and the structure claimed to be proved by uplc/ms. The danger of using this method is illustrated by the fact that a molecular formula as C15H10O5 presents more than 470 compounds. This number will increase by more complex molecules. A method for a decent identification by ms has been described by Reisdorph et al. (Reisdorph, Walmsley et al. 2020). It is strongly suggested that to get a level 1 identification an authentic compound has to be used for comparison. No such ateempt has been made in this manuscript. I therefore have to suggest rejection because of missing proof for the suggested structures.

Reisdorph, N. A., et al. (2020). "A perspective and framework for developing sample type specific databases for LC/MS-based clinical metabolomics." Metabolites 10(1): 8.

No comments

Author Response

 Comments 1: Some compounds have been detected and the structure claimed to be proved by uplc/ms. The danger of using this method is illustrated by the fact that a molecular formula as C15H10O5 presents more than 470 compounds. This number will increase by more complex molecules. A method for a decent identification by ms has been described by Reisdorph et al. (Reisdorph, Walmsley et al. 2020). It is strongly suggested that to get a level 1 identification an authentic compound has to be used for comparison. No such ateempt has been made in this manuscript. I therefore have to suggest rejection because of missing proof for the suggested structures.

Reisdorph, N. A., et al. (2020). "A perspective and framework for developing sample type specific databases for LC/MS-based clinical metabolomics." Metabolites 10(1): 8.

Response 1: Dear reviewer, thank you for your comments. Our level of identification has been validated only for the compounds marked with an * in the results table, indicating that confirmation was performed through comparison with authentic commercial standards. For the other proposed compounds, the identification was based on the analysis of the exact mass of the molecular ion in comparison to the molecular formula, ensuring a mass error of <5 ppm. Additionally, we analyzed the fragmentation pattern and isotopic distribution. Furthermore, we conducted analyses by comparing our mass spectra with databases using the Progenesis QI software. Altogether, this approach gives us a level 2 (high) identification with putative annotations. This strengthens the reliability of our identifications, though we acknowledge that level 1 identification was not achieved for all compounds. But we agree with your comment, in the document we omitted the meaning of *, so we corrected the legend of Table 2 and added the sentence: *Identification confirmed with commercial standards. Please check the change in the document.

Revised manuscript:

Page 9, Legend: *Identification confirmed with commercial standards.

Reviewer 2 Report

Authors studied the characterization of extractable and non-extractable phenols and betalains in berrycactus (Myrtillocactus geometrizans) and its chemoprotective effect in early stage of colon cancer in vivo. They obtained novel substances from the fruit bodies of Myrtillocactus geometrizans. Authors demonstrated that extractable and non-extractable polyphenols of berrycactus exhibits antioxidant activities and chemoprotective effect. Berrycactus and berrycactus residue also exhibits chemoprotective effect in AMO/DSS animal model. The manuscript is a novel study. However, a few doubts need to be illustrated.

Major comments

1.     In Introduction (line 68), why authors studied the phytochemical characterization of “betalain”? Authors should explain in the Introduction section.

2.     Why betalain did not appear in the study of chemoprotective effect?

3.     In the in vivo experimental design, why authors did not pretreatment of BC or BRC at 1-3 weeks, then treat AMO/DSS and euthanization at 16 weeks. This experimental design is particularly useful for assessing preventive efficacy.

4.     In Figure 2, why the numbers of ACF in both BC+AMO/DSS and BCR+AMO/DSS groups are significantly more than AMO/DSS group in 2C? Authors should discuss.

Minor comments

1.     In Abstract (line 19 and line 24), the “AMO/DSS”, “UPLC-QTOF-MSE”, and “ORAC” are first appearance on the manuscript. An acronym must be expanded/defined at first mention in both the abstract and in the main text. Subsequent usage should mention the acronym alone.

2.     In Abstract (line 22), the “H2SO4and” should revise to “H2SO4 and”.

3.     In Abstract (line 23), the “flavonoid” should revise to “flavonoids”.

4.     In Introduction (line 68), the “NPP” are first appearance on the manuscript. An acronym must be expanded/defined at first mention in both the abstract and in the main text. Subsequent usage should mention the acronym alone.

5.     Page 4, line 174, “ACF” are first appearance on the manuscript. An acronym must be expanded/defined at first mention in both the abstract and in the main text. Subsequent usage should mention the acronym alone.

Author Response

Reviewer 2

Authors studied the characterization of extractable and non-extractable phenols and betalains in berrycactus (Myrtillocactus geometrizans) and its chemoprotective effect in early stage of colon cancer in vivo. They obtained novel substances from the fruit bodies of Myrtillocactus geometrizans. Authors demonstrated that extractable and non-extractable polyphenols of berrycactus exhibits antioxidant activities and chemoprotective effect. Berrycactus and berrycactus residue also exhibits chemoprotective effect in AMO/DSS animal model. The manuscript is a novel study. However, a few doubts need to be illustrated.

Major comments

Comments 1. In Introduction (line 68), why authors studied the phytochemical characterization of “betalain”? Authors should explain in the Introduction section.

Response 1: We enormously appreciate the reviewer’s recommendations to make this article more readable and proper for publication in Antioxidants. Betalains are characterized by being the natural pigment of berrycactus, like anthocyanins in other non-cactus species, as previously reported by Guzmán-Maldonado et al [1] in fresh fruits and by Sánchez-Recillas et al. [2] after an in vitro digestion process of berrycactus. The presence of betalains is mentioned in the introduction (lines 50-55), which justifies the inclusion of the search for these compounds in the phytochemical characterization.

  1. Guzmán-Maldonado, S.H.; Herrera-Hernández, G.; Hernández-López, D.; Reynoso-Camacho, R.; Guzmán-Tovar, A.; Vaillant, F.; Brat, P. Physicochemical, nutritional and functional characteristics of two underutilised fruit cactus species (Myrtillocactus) produced in central Mexico. Food Chem 2010, 121, 381–386.
  2. Sánchez-Recillas, E.; Campos-Vega, R.; Pérez-Ramírez, I.F.; Luzardo-Ocampo, I.; Cuéllar-Núñez, M.L.; Vergara-Castañeda, H.A. Garambullo (Myrtillocactus geometrizans): effect of in vitro gastrointestinal digestion on the bioaccessibility and antioxidant capacity of phytochemicals. Food Funct 2022, 13, 4699–4713.

Comments 2. Why betalain did not appear in the study of chemoprotective effect?

Response 2: Thanks for the observation. Physicochemical characterization of berrycactus by UPLC-QTOF-MSE revealed the presence of 4 betalains in the extractable polyphenol fraction, while they could not be detected in the non-extractable polyphenol fraction. This could be due to structural changes during their passage through the chemical and mechanical conditions of the gastrointestinal tract, which do not allow their identification. Betalains are polar, therefore the results show that they remain in the extractable polyphenol fraction. From a physiological point of view, it is known that betalains are absorbed in the small intestine and enter the liver via the bloodstream, where they undergo further metabolism. Therefore, it is reasonable to assume that betalains do not reach the large intestine, or at least a large part of their concentration, and that they could contribute to the protective effect of berry cactus via other organs such as the liver, but this last statement was not confirmed in the proposed experimental design.

Comments 3. In the in vivo experimental design, why authors did not pretreatment of BC or BRC at 1-3 weeks, then treat AMO/DSS and euthanization at 16 weeks. This experimental design is particularly useful for assessing preventive efficacy.

Response 3: We greatly appreciate the reviewer’s comment. The treatment with BC or BCR was administered during the 16 weeks of the experimental design, and the AOM/DSS administration occurred in the 3rd and 4th weeks. Therefore, the experimental design corresponds to a preventive evaluation by administering the treatments before, during and after the AOM/DSS. We have modified the Figure 1 for better understanding, as well as a small change to the text for the same purpose.

Revised manuscript:

Page 4, Line 162. …(weeks 3 and 4), and at the same time a 7-day cycle…

Page 4, Line 171 Figure 1.

Comments 4. In Figure 2, why the numbers of ACF in both BC+AMO/DSS and BCR+AMO/DSS groups are significantly more than AMO/DSS group in 2C? Authors should discuss.

Response 4: We appreciate the reviewer’s comment. Multiplicity of ACF refers to the number of crypts per focus found per treatment group. Multiplicity refers to the potential risk of an ACF progressing to form an adenoma or adenocarcinoma. BC+AOM/DSS and BCR+AOM/DSS had a higher number of ACFs with lower multiplicity, i.e. 2 crypts per focus, while the AOM/DSS group had a lower number of ACFs with low multiplicity and a higher number of ACFs with higher multiplicity. This means a higher potential for the development of malignant lesions in the colon.

Revised manuscript:

 Page 11. BC+AOM/DSS and BCR+AOM/DSS showed a higher number of ACFs with low multiplicity (2C), while AOM/DSS induced ACFs with higher multiplicity (3C and 4C), indicating a higher potential for development into malignant lesions in the colon. In addition, significant differences were found…

Minor comments

Comments 1. In Abstract (line 19 and line 24), the “AMO/DSS”, “UPLC-QTOF-MSE”, and “ORAC” are first appearance on the manuscript. An acronym must be expanded/defined at first mention in both the abstract and in the main text. Subsequent usage should mention the acronym alone.

Response 1: Thanks. You are right, the acronym must be defined at the first mention, but to integrate the entire experimental setup and results into a 200-word abstract, we felt it necessary to omit the definition of acronyms. In the Materials and Methods section, where the capacitance techniques used are described, the definitions of the acronyms mentioned have been added.

Comments 2. In Abstract (line 22), the “H2SO4and” should revise to “H2SO4 and”.

Response 2: We have corrected the text as you suggested.

Revised manuscript:

 Page 1, line 23. … with H2SO4 and HCl… 

Comments 3. In Abstract (line 23), the “flavonoid” should revise to “flavonoids”.

Response 3: Thanks. We have changed the text of the abstract, so that the term "flavonoid" is now used correctly.

Revised manuscript:

Page 1, line 24. … total phenolics and flavonoid,…

Comments 4. In Introduction (line 68), the “NPP” are first appearance on the manuscript. An acronym must be expanded/defined at first mention in both the abstract and in the main text. Subsequent usage should mention the acronym alone.

Response 4: Thanks for the observation. We have made a mistake. The correct acronym is NEP, which was previously referred to as non-extractable polyphenols. Therefore, the text has been changed from NNP to NEP.

Revised manuscript.

Page 2, line 68. … extractable polyphenols (EP), NEP and betalains…

Comments 5. Page 4, line 174, “ACF” are first appearance on the manuscript. An acronym must be expanded/defined at first mention in both the abstract and in the main text. Subsequent usage should mention the acronym alone.

Response 5: Thanks. We have corrected the sentence.

Revised manuscript.

Page 5, line 182. …development of aberrant crypt foci (ACF) by staining…

Round 2

Reviewer 1 Report

The authors acknowledge that the identification of compounds based solely on MS is dubious. To express this, they graduate identification. I disagree with this approach. When reporting the presence of a compound you collect all data necessary for an unequivocal identification, or you do not report the presence of the compound. Today you may run NMR on microgram amounts of compounds. Consequently, no excuse exists for publishing “likely” structures.  

No comments

Author Response

Comments 1: The authors acknowledge that the identification of compounds based solely on MS is dubious. To express this, they graduate identification. I disagree with this approach. When reporting the presence of a compound you collect all data necessary for an unequivocal identification, or you do not report the presence of the compound. Today you may run NMR on microgram amounts of compounds. Consequently, no excuse exists for publishing “likely” structures.  

Response 1: We thank the reviewer for his comments. The reviewer is right. The best way to report the presence of a compound would be to use techniques such as NMR. However, this is a method that is not available to us. UPLC-QTOF-MS is a sensitive and selective technique where the observed mass is very similar to the theoretical mass, so there is a high probability that the compound is present. We consider that the analysis of the exact mass of the molecular ion, the comparison of the mass spectra with the database used (Progenesis QI software) and the identification level (2) provide sufficient reliability in the identification. Even today, this technique is still widely used in research to identify compounds (1,2,3,4. However, it cannot be ruled out that more accurate identification of these compounds using other identification methods will be sought in the future.

  1. Fernández-Ochoa, Á.; Younis, I.Y.; Arafa, R.K.; Cádiz-Gurrea, M.d.l.L.; Leyva-Jiménez, F.J.; Segura Carretero, A.; Mohsen, E.; Saber, F.R. Metabolite Profiling of Colvillea racemosavia UPLC-ESI-QTOF-MS Analysis in Correlation to the In Vitro Antioxidant and Cytotoxic Potential against A549 Non-Small Cell Lung Cancer Cell Line. Plants 202413, 976. https://doi.org/10.3390/plants13070976
  2. Kim, Y.J.; Jang, S.; Hwang, Y.-H. Qualitative and Quantitative Analysis of Phytochemicals in Sayeok-Tang via UPLC-Q-Orbitrap-MS and UPLC-TQ-MS/MS. Pharmaceuticals202417, 1130. https://doi.org/10.3390/ph17091130
  3. Oh, H.-B.; Jeong, D.-E.; Lee, D.-E.; Yoo, J.-H.; Kim, Y.-S.; Kim, T.-Y. Structural Identification of Ginsenoside Based on UPLC-QTOF-MS of Black Ginseng (Panax GinsengA. Mayer). Metabolites202414, 62. https://doi.org/10.3390/metabo14010062
  4. Jiang, F.; Li, M.; Huang, L.; Wang, H.; Bai, Z.; Niu, L.; Zhang, Y. Metabolite Profiling and Biological Activity Assessment of Paeonia ostiiAnthers and Pollen Using UPLC-QTOF-MS.  J. Mol. Sci. 202425, 5462. https://doi.org/10.3390/ijms25105462

Reviewer 2 Report

Accept in present form..

Accept in present form.

Author Response

Comments 1: Accept in present form.

Response 1: We greatly appreciate the reviewer’s comment.
